# Non-Intrusive Detection of Occupants' On/Off Behaviours of Residential Air Conditioning

**Tetsushi Ono, Aya Hagishima ***  **and Jun Tanimoto**

Interdisciplinary Graduate School of Science Engineering (IGSES), Kyushu University Kasuga-koen 6-1, Kasuga-shi, Fukuoka 816-8580, Japan

\* Correspondence: ayahagishima@kyudai.jp

**Abstract:** Understanding occupants' behaviours (OBs) of heating and cooling use in dwellings is essential for effectively promoting occupants' behavioural change for energy saving and achieving efficient demand response operation. Thus, intensive research has been conducted on data collection, statistical analysis, and modelling of OBs. However, the majority of smart metres currently deployed worldwide monitor only the total household consumption rather than appliance-level load. Therefore, estimating the turn-on/off state of specific home appliances from the measured household total electricity referred to as non-intrusive load monitoring (NILM), has gained research attention. However, the current NILM methods overlook the specific features of inverter-controlled heat pumps (IHPs) used for space heating/cooling; thus, they are unsuitable for detecting OBs. This study presents a rule-based method for identifying the occupants' intended operation states of IHPs based on a statistical analysis of load data monitored at 423 dwellings. This method detects the state of IHPs by subtracting the power of sequential-operation appliances other than IHPs from the total household power. Three time-series characteristics, including the durations of power-on/off states and power differences between power-off/on states, were used for this purpose. The performance of the proposed method was validated, indicating an F-score of 0.834.

**Keywords:** occupants' behaviour; residential buildings; HVAC system; rule-based method

## 1. Introduction

With growing concerns about global climate change in recent years, reducing greenhouse gas emissions has become an urgent issue. The building sector accounted for approximately 36% of the global energy demand and 37% of energy-related $CO_2$ emissions in 2020 [1]; thus, reducing energy demand through behavioural change and improving energy efficiency, as well as increasing renewable energy penetration, are crucial. Furthermore, increasing demand flexibility has gained importance to ensure a supply–demand balance for stable power supply operations, consistent with the increase in renewable energy sources.

Heating, ventilation, and air conditioning (HVAC) play a crucial role in reducing the building energy demand because of their extensive contribution to the total demand. Gonzalez-Torres et al. [2] reported that HVAC systems are the most consumed service worldwide (38%) in both residential (32%) and tertiary (47%) sectors. Since the primary function of HVAC is to establish healthy and thermally comfortable indoor spaces, the demand for HVAC largely depends on the building envelope performance, energy efficiency of HVAC, and occupants' behaviours (OBs), such as set-point temperature and duration of use [3,4]. The roadmap toward net zero (NZE) presented by the International Energy Agency emphasised the importance of behavioural changes for emission reduction in buildings in the NZE scenario [5].

Demand response (DR) is a novel approach to demand flexibility that refers to methodologies to encourage consumer response to take energy-saving actions or shift the time of

energy use through various schemes, such as time-of-use electricity pricing [6–8], incentive payments designed to induce lower electricity use at peak hours [9], and smart metering systems of electricity consumption, including customer feedback functions [10]. Home-energy management systems are widely used for providing feedback and smart metering through in-home displays [11,12]. Ehnhardt-Martinez et al. [13] summarised the results of 36 past studies on DR and established that electricity use was reduced by 8.0% to 12.0% through the behavioural changes triggered by metering-based advice.

Understanding the OBs on heating and cooling use in dwellings is crucial for designing effective schemes to promote behavioural change and achieve appropriate performance in the actual DR operation. In particular, the energy consumption of space heating/cooling in the residential sector highly depends on the occupants' various behavioural patterns, such as their living schedules, thermal preferences, and personal habits. Therefore, intensive research has been conducted on the data collection, statistical analysis, and modelling of OBs in relation to stochastic building energy simulations.

To monitor the usage status of home appliances, for example, when appliances are running (hereinafter called the 'turn-on-state') or out-of-operation with no electricity use (hereinafter called the 'turn-off-state'), installing sensors to target appliances is a straightforward method. However, the majority of smart metres currently deployed in the residential sector can monitor only the total household consumption. Multiple-appliance metering is generally expensive and may cause privacy concerns for customers. Therefore, technologies for estimating the turn-on/off state of each or specific home appliance from the time-series patterns of measured household total electricity demand data, called non-intrusive load monitoring (NILM), have gained popularity among researchers for decades [14]. Past studies on NILM can be classified into two types—high sampling frequency from several tens of kHz to 1 Hz and low sampling frequency from 1 Hz to 1/60 Hz—Based on the frequency of the monitored load data used [15].

Most studies on NILM from the 1990s and the 2000s are based on high-frequency sampled data. The advantage of using high-frequency data is that they can capture electrical characteristics, such as distortions and harmonics of current and voltage waveforms, when switching home appliances. For example, Leeb et al. [16] proposed a method for estimating the switching of each home appliance based on the spectral envelope created by the Fourier transform. Murata et al. [17] presented a method for utilising the harmonic current and phase data to estimate the electricity consumption of each device at each time step.

In the 2010s, with the spread of smart metering, open datasets of the electricity demand of dwellings measured at low sampling frequencies, such as REDD [18] and UK-DALE [19], were released. This accelerated NILM research using low sampling frequency data. Such NILM studies use sparse coding [20], hidden Markov models [21,22], and methods based on feature extraction of time-series patterns [23]. In addition, modern machine learning technologies, such as deep neural networks [24,25], Boltzmann machines [26], and Bayesian classifiers [27], have been applied to NILM.

Unlike usual home appliances, the usage trends of air conditioners (ACs) for space cooling or heat pumps (HPs) for both heating and cooling have distinct seasonal characteristics, and they significantly contribute to the annual peak demand. Therefore, NILM algorithms, specifically for ACs or HPs, are being developed to improve accuracy. For example, Perez et al. [28] proposed an estimation method using k-means and validated their model using an open dataset measured at Pecan Street Inc. in the U.S. [29]. Su et al. [30] applied a support vector machine and confirmed the estimation performance using the Pecan Street dataset. Inoue et al. [27] adopted averaged one-dependence estimation (AODE), which is a type of Bayesian classifier, and tested the performance of the model based on the experimental data of electricity consumption of fixed-frequency HPs. In general, compressors of fixed-frequency HPs mostly operate at a predetermined speed; thus, electricity consumption tends to be high [29]. The basic ideas of these studies have some similarities; when high-power consumption continues for a certain duration, ACs

or HPs are likely to be in the turn-on state. The assumption adopted in these studies was validated using the demand data of dwellings equipped with fixed-frequency HPs.

Residential ACs or HPs that assist in controlling the speed of the compressor for optimum operation have recently improved energy efficiency significantly compared to conventional fixed-frequency ones, and they are rapidly replacing conventional fixed-frequency ACs and HPs worldwide [30]. For example, inverter-controlled HPs (IHP) had a 100% share in the Japanese residential market by 2020, and the same is expected to increase continuously worldwide. The characteristics of the load patterns of IHPs are different from those of the aforementioned conventional methods [31–33]. First, there are periods with no electricity consumed despite HPs being in operation, as their compressors repeatedly rotate and stop when the thermal load is small. Second, even during the HP-running periods, the electricity consumption is typically low based on the thermal load by reducing the compressor rotation speed. Such characteristics of IHPs mean that NILM is more challenging than conventional fixed-frequency HPs. Moreover, the characteristics of IHPs, in which the power-on/off state determined by electricity consumption does not necessarily correspond to the occupants' intended turning-on/off behaviours, cause problems in studies aimed at detecting occupants' heating and cooling use behaviour [34,35]. Ono et al. [36] proposed an algorithm for identifying the time of occupants' switching-on/off behaviour taken from the time-series electricity data of IHPs, based on OB monitoring and metered electricity data of HPs. However, to the best of our knowledge, no studies on NILM have considered the discrepancy between the occupant's intended IHP switching behaviour and the power-on/off status by monitoring household electricity demand.

This study proposes a rule-based method for estimating the intended behaviour of occupants to turn on/off HPs applicable to current IHPs. First, we conducted a statistical analysis of the appliance-level electricity consumption dataset of 586 dwellings using IHPs to identify the time-series characteristics of the electricity consumed by various home appliances. Subsequently, we proposed a rule-based algorithm for estimating the turn-on/off states of the HPs. Modern machine learning methods can be considered another option; however, they have the disadvantage of having numerous parameters and a large computational load during both training and inference procedures. However, the computational load of rule-based methods is generally low. This facilitates the transmission of energy-saving information to occupants using limited computing resources. The proposed method can be applied to a large amount of existing total household electricity consumption data to generate a substantial pool of data on the operating states of HPs. The method is expected to be of considerable use to researchers in elucidating the OB characteristics as well as in appropriate OB modelling. Furthermore, the proposed method can be used to issue power-saving reminders based on the operational status of the HPs to implement a DR programme for households.

The remainder of the paper is organised as follows (Figure 1). Section 2 presents an overview of the electricity demand dataset used in the analysis. Section 3 discusses the extracted time-series characteristics of the electricity consumption of the IHPs based on the dataset. Section 4 proposes a rule-based method for estimating the turn-on/off states of the IHPs. Section 5 discusses the accuracy of the proposed method. Finally, Section 6 presents the conclusions.

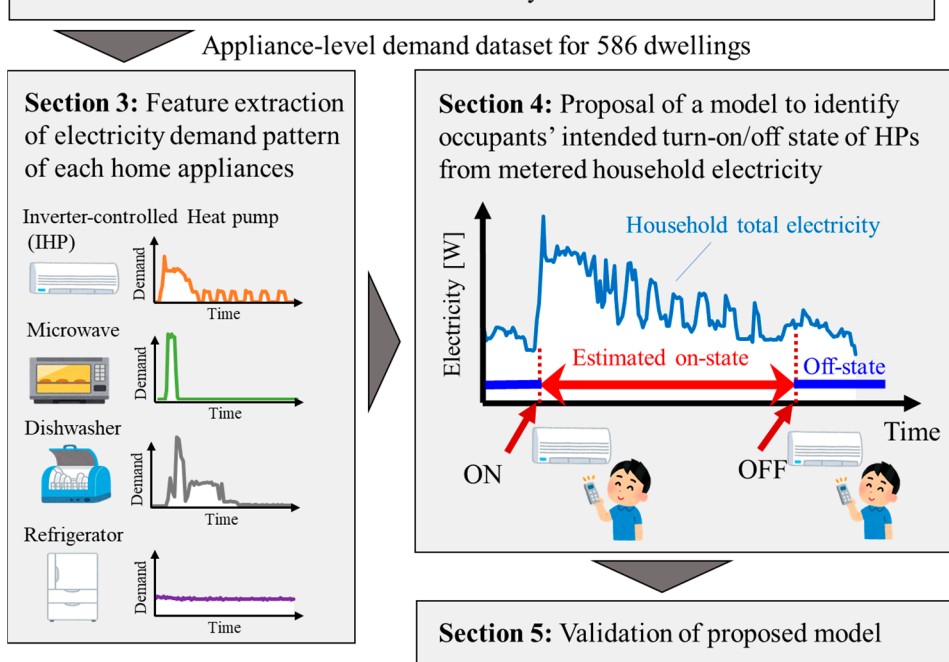

**Figure 1.** Overall research process in this study.

## 2. Summary of Dataset

### 2.1. Target Dwellings

We used a one-year dataset of the electricity demand measured at 586 dwellings from 1 January to 31 December 2013. All of the surveyed dwellings were located in the same residential building of a 20-story, large-scale housing complex in Settsu City, Osaka Prefecture, Japan. A summary of the target residential building is provided in Table 1.

**Table 1.** Measurement target housing complex.

| Location | Settsu City, Osaka, Japan |
| --- | --- |
| Number of stories | 20 |
| Completion date | January 2011 |
| Structure | Reinforced concrete structure |
| Building envelopes | External walls: internal insulation with air layer, U-value 0.411 W/(m$^2$ K) <br> Windows: Low-E double-glazing |
| Number of dwellings by layout (average floor area) | Total 586 dwellings <br> 38 dwellings: 2 bedrooms + LDK [a] (55.1 m$^2$) <br> 391 dwellings: 3 bedrooms + LDK [a] (71.2 m$^2$) <br> 157 dwellings: 4 bedrooms + LDK [a] (83.6 m$^2$) |

[a] LDK refers to a unified space used for living rooms, dining rooms, and kitchens.

The electricity consumption of the total household and appliance level, including HPs, refrigerators, washing machines, lighting, and outlet of each room, was measured for each dwelling. Moreover, the accumulation of electricity consumption (Wh) was measured at an interval of 1 min. The measurement accuracy was ±5%.

IHPs were installed in the living rooms of all dwellings when the construction was completed in 2011. The specifications of the IHPs in the living rooms are summarised in Table 2. Five types of IHPs with different outputs manufactured by the same company were installed according to the room area. The annual performance factor of the installed IHPs, which indicates energy efficiency, ranged from 4.7–6.7. Conversely, the HPs in other

rooms (e.g., bedrooms) were installed by the residents themselves; thus, the models are unknown. In addition, the same dishwasher models were installed in all dwellings.

**Table 2.** Specification of inverter-type heat pumps in living rooms.

| Room Floor Area [m²] | Cooling Capacity [kW] | Heating Capacity [kW] | Annual Performance Factor |
| :---: | :---: | :---: | :---: |
| 18 | 2.8 (0.6–4.2) | 3.2 (0.6–7.9) * | 6.7 |
| 26 | 4.0 (0.6–5.4) | 5.0 (0.6–10.4) * | 6.3 |
| 33 | 5.0 (0.6–5.9) | 6.0 (0.6–10.4) * | 5.7 |
| 36 | 6.3 (0.6–6.5) | 7.1 (0.6–10.4) * | 5.1 |
| 42 | 7.1 (0.6–7.3) | 7.5 (0.6–10.4) * | 4.7 |

* () in cooling/heating capacities indicates mean, minimum, and maximum values.

### 2.2. Breakdown of Electricity Demand by Use

Figure 2 shows the seasonal variations in the average electricity demand per dwelling. The error bars indicate the standard deviation of the variations in total electricity demand among the dwellings. The electricity demand of the HPs for living rooms and other rooms is shown separately. Although the electricity demand for the outlets and lights was separately measured in living rooms, the total electricity demand for the outlets and lights was measured in other rooms (e.g., bedrooms). The electricity consumption of outlets in living rooms, refrigerators, and HPs in living rooms was high, accounting for 25.0%, 18.3%, and 9.5% of the total annual demand, respectively (Figure 2). The electricity consumption of HPs in bedrooms was considerably lower than that in living rooms. This can be attributed to the smaller floor area and shorter time of HP use for bedrooms than for living rooms. Considering time variations, as expected, HPs exhibited an evident seasonal trend with large winter and summer consumption. For example, the total electricity consumptions in August and February were approximately 1.6 and 1.4 times that in May, respectively. Although the values of summer were larger than those of winter, the heating degree days of the target year were larger than the cooling ones. This is caused by the combined usage of the gas floor heating system and HPs in the living rooms [4]. In addition to HPs, the electricity consumption by refrigerators shows a weak annual cycle, with summer consumption being 77% greater than in winter. The variations in total electricity consumption among households are large, with standard deviations ranging from 0.66 to 0.83 times the sample mean.

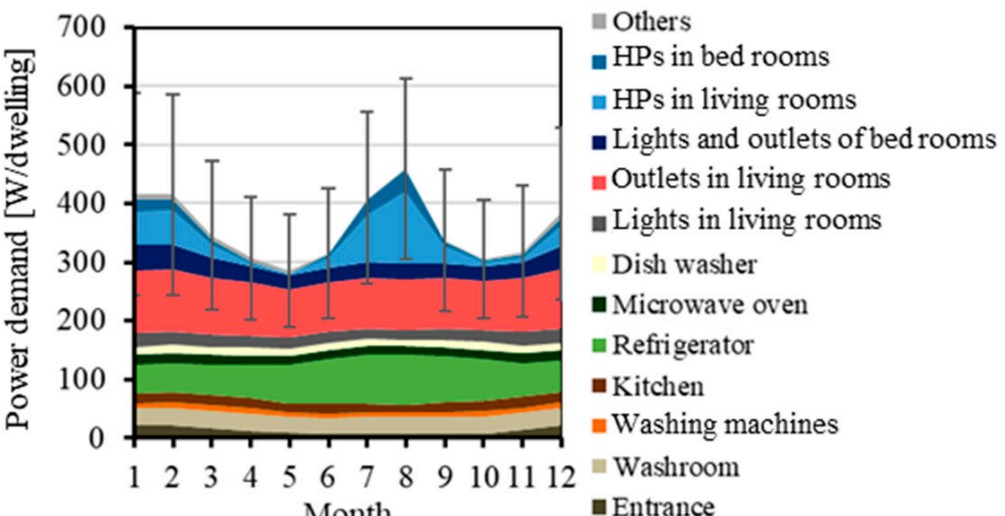

**Figure 2.** Seasonal variations in electricity consumption by use.

Figure 3 shows the daily variations in electricity consumption during winter (December–March), mid-season (April, May, October, and November), and summer (June–September). The total electricity demand was minimum at approximately 4:00 h, increased at 7:00 h, slightly decreased during the daytime (10:00–16:00 h), and reached its maximum at approximately 19:00 h during all three seasons. This diurnal variation is attributed to most of the items, except for refrigerators, such as lights, HPs, and dishwashers. Comparing the trends in HPs' electricity between summer and winter, it is noted that the values in winter are larger at around 6:00–7:00 h in the morning when the outside temperatures are lower, resulting in sharper peak demand in total household summers. In contrast, summers show larger values than winters during the daytime when the cooling load is higher owing to solar radiation and high outdoor temperatures. The variation in total household electricity among households was the most significant in winter. This is partly because the dwellings had both gas-driven underfloor heating and electricity-driven HPs for heating living rooms, and there were differences in terms of which of the two heating systems was used more frequently in each dwelling.

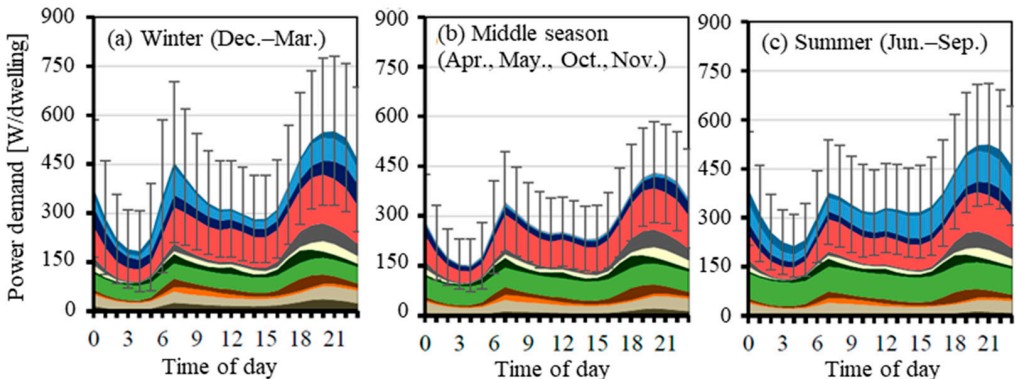

**Figure 3.** Daily variations in electricity consumption by use.

*2.3. Electricity Demand Patterns of IHPs*

Figure 4 shows two examples of the observed time-series electricity demand patterns of HPs made by the same manufacturer with the same specifications installed in living rooms. Since the measurement dates of patterns 1 and 2 were August and September 2013, respectively, the HPs were supposed to be used for cooling. Both plots indicate that periods with no electricity consumption appeared intermittently. Furthermore, the timing and frequency of such intermittent zero-consumption periods differed between the two patterns. This regular intermittency was caused by the inverter control of the HPs and not by the occupants' switching-on action. The difference between the plots can be attributed to the difference in the thermal load that the HPs have to tackle for establishing the set-point temperature in each room. This tendency suggests the diversity of the demand patterns of HPs.

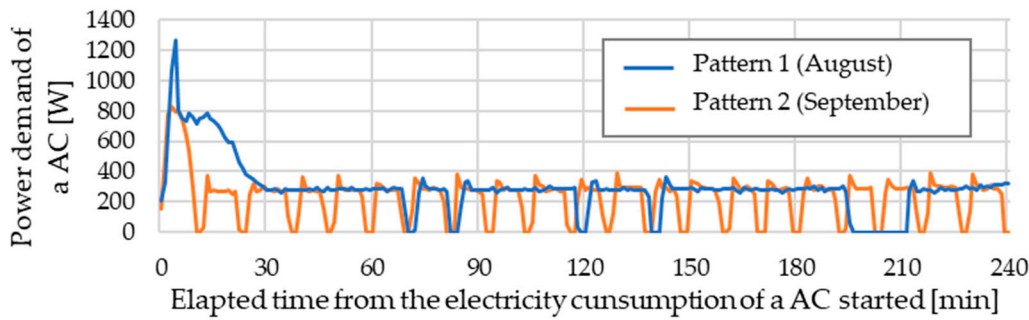

**Figure 4.** Examples of electricity demand patterns on an IHP in living rooms.

Notably, the HP switch was likely to remain on for occupants during the 240 min period, as shown in Figure 4. Hence, the on/off states of HPs determined by zero-consumption periods are not necessarily consistent with the actions of occupants to turn on/off HPs. Hereafter, the on/off state of an HP determined by an occupant's behaviour is referred to as the turn-on/off state, and the state with an HP power consumption of zero or non-zero is referred to as the power-on/off state.

## 3. Characteristics of Electricity Consumption Pattern

### 3.1. Variables to Characterise the Load Patterns

To quantify the time-series characteristics of the electricity consumption patterns of various uses, the following variables were defined:

First, $\Delta P$, change in electricity consumption (W) can be defined as follows:

$$\Delta P(t) = P(t+1) - P(t-1) \tag{1}$$

$\Delta P$ is $\Delta$ typically defined as the difference of one time step (e.g., $t$ and $t-1$); however, in this study, we defined it as the lag between two time steps (i.e., $t+1$ and $t-1$) to avoid the over-detection of the start of appliance operation caused by the cumulative power consumption measured for events initiated in the middle of a 1 min measurement interval. Using $\Delta P(t)$ and threshold $\alpha$, the time of ith event with a large lag of consumption is determined as follows:

$$t_{on,i} = t; \text{ if } \Delta P(t) > \alpha \tag{2}$$

$$t_{off,i} = t; \text{ if } \Delta P(t) < -\alpha \tag{3}$$

In a time sequence, $t_{on,i}$ and $t_{off,i}$ are expected to alternate. When the same signal of either $t_{on,i}$ or $t_{off,i}$ is detected consecutively, only the signal that appears last is adopted. Based on $t_{on,i}$ and $t_{off,i}$ defined above, the time intervals between the two are defined as follows:

$$\Delta t_{on,i} = t_{on,i} - t_{off,i} \tag{4}$$

$$\Delta t_{off,I} = t_{off,i} - t_{on,i+1} \tag{5}$$

As the equations indicate, $\Delta t_{on,i}$ and $\Delta t_{off,i}$ are the time intervals between the time when the measured power consumption significantly increased/decreased and the next time when it decreased/increased, respectively.

In addition, $\Delta ON_i$ and $\Delta OFF_i$ can be defined as follows:

$$\Delta ON_i = \Delta P(t_{on,i}) \tag{6}$$

$$\Delta OFF_i = \Delta P(t_{off,i}) \tag{7}$$

$\Delta ON_i$ and $\Delta OFF_i$ are the differences in electricity consumption when the transition from a power-off state to a power-on state or vice versa occurs.

Figure 5 shows the definitions of the above variables, turn-on/off state, and power-on/off state. The turn-on/off state was changed at the time of the occupant's action. Meanwhile, the power-on/off states were determined by the threshold of the HP electricity. The period of $\Delta t_{on,i}$ corresponded to a power-on state where power is consumed at some level, and the period of $\Delta t_{off,i}$ corresponded to the power-off state with approximately zero power consumption lower than the threshold.

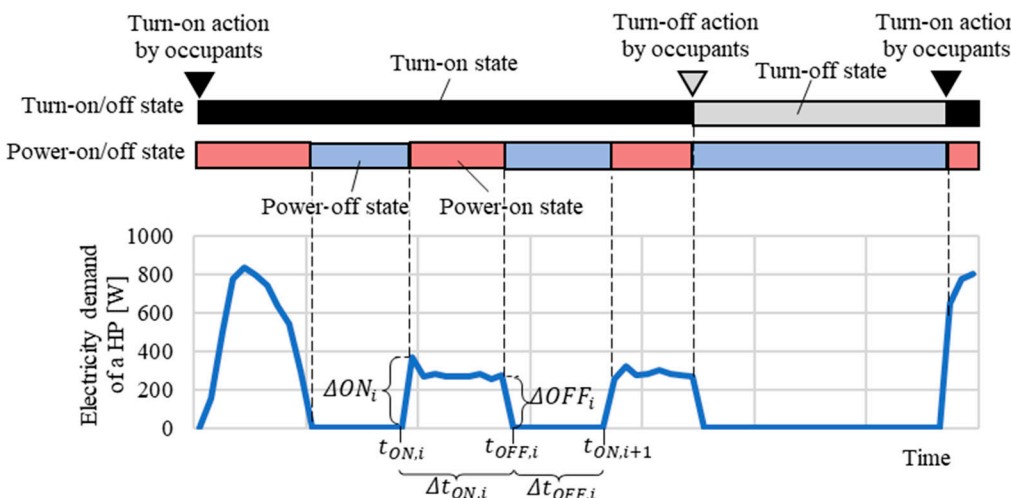

**Figure 5.** Illustrations of terminologies.

### 3.2. Time-Series Characteristics of Electricity Consumption of Appliances

We calculated the variables determined in the previous section using the electricity consumption data of HPs and other major appliances. Figure 6 shows the joint probability (JP) distribution of $\Delta$ON and $\Delta t_{on}$ of eight major appliances. Notably, (g) indicates the total electricity demand of outlets and lights in all bedrooms, and (h) includes all the HPs installed in both living rooms and bedrooms.

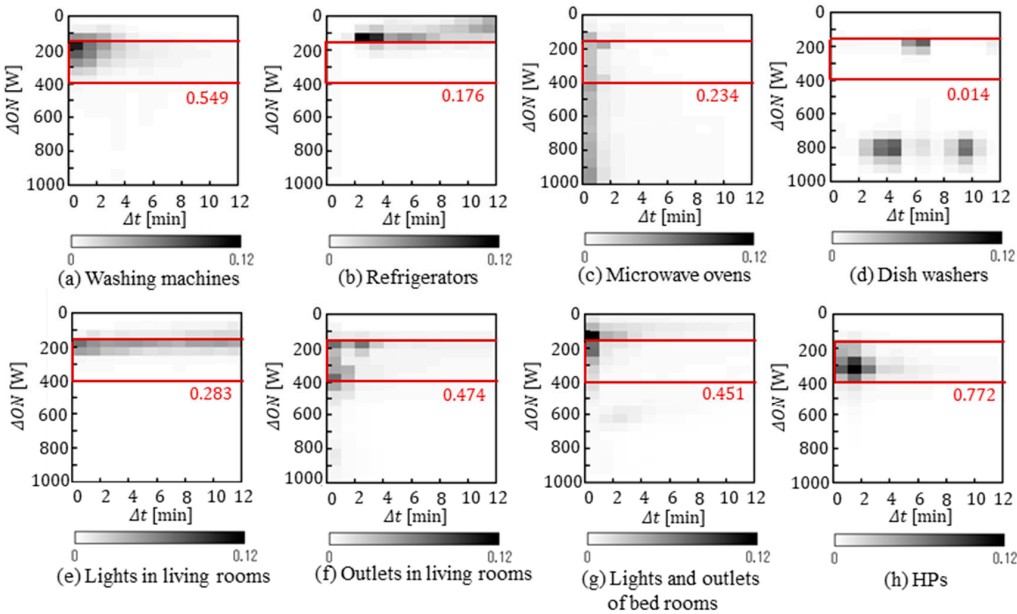

**Figure 6.** Joint probability distribution of $\Delta$ON and $\Delta t_{on}$ in the measured electricity consumption data of eight major appliances. Numbers marked in red in the graphs indicate the conditional probabilities within the range of 150 < $\Delta$ON < 400 W and $\Delta t_{ON}$ < 15 min.

The JP distribution of (h) HPs showed high probability when $\Delta$ON was between 150 W and 400 W and $\Delta t_{on}$ was less than 6 min. This result is consistent with the frequent repetition of power-on/off states dominated by the inverter control shown in Figure 4.

The JP distribution of (a) washing machines had a high probability when $\Delta$ON was between 100 W and 350 W and $\Delta t_{on}$ was below 6 min. Washing machines are generally operated according to a specific sequence (e.g., washing → dehydration); thus, the electricity demand patterns of such appliances (hereinafter referred to as 'sequential operation

appliances') may have specific features, being less diverse. The JP distribution of another sequential operation appliance, namely (d) dishwashers, indicates a unique tendency compared to the other appliances, having three major peaks. This may be attributed to the fact that the same dishwashers made by the same manufacturer were equipped in all the dwellings.

Conversely, for (b) refrigerators and (e) lighting, the JP of ΔON was high in the conditions of less than 250 W but widely scattered against $\Delta t_{on}$. For (c) microwave ovens, the JP was mostly observed for $\Delta t_{on}$ less than 10 min but widely scattered against ΔON. Different JP distributions for the appliances shown in Figure 6 imply the effectiveness of ΔON and $\Delta t_{on}$ in identifying the power-on/off state of specific appliances from the total power consumption patterns.

The conditional probability of HPs for the condition of ΔON ranging from 150 to 400 W and $\Delta t_{on}$ ranging from 1 to 15 min was the largest at 0.772, followed by washing machines, outlets in living rooms, and outlets and lights of bedrooms, all of which accounting for more than 0.4. In contrast, the probabilities of other appliances were lower than 0.3. The assumption that HPs are turned on when the total household electricity consumption satisfies the conditions of 150 W < ΔON < 400 W and $\Delta t_{on}$ < 15 min might be acceptable if we can exclude the influence of other appliances, such as washing machines, with relatively high conditional probability for the same ranges of ΔON and $\Delta t_{on}$.

In addition to ΔON and $\Delta t_{on}$, the presence or absence of a sequence of intermittent power-on/off states shown in Figure 4 may be a key for identifying the turn-on/off state of HPs. Accordingly, we considered power-off states of ΔOFF < 15 min as part of a turn-on-state and computed the number of power-on states connected by power-off states of ΔOFF < 15 min (denoted as NPOS) for each appliance.

Figure 7 shows the cumulative probability density (CPD) of NPOS for the eight major appliances. The CPD reached 0.95 or higher at NPOS = 3 for all appliances except for HPs and washing machines. The CPD of washing machines also rapidly increased at NPOS = 3, and the difference from that of HPs became large, reaching approximately 0.98 at NPOS = 6. Conversely, the CPD of HPs was considerably lower than that of the others, reaching 0.95 at NPOS = 30 and 0.98 at NPOS = 50. This can be used to differentiate the turn-on/off of HPs from other appliances, in addition to the abovementioned features of ΔON and $\Delta t_{on}$.

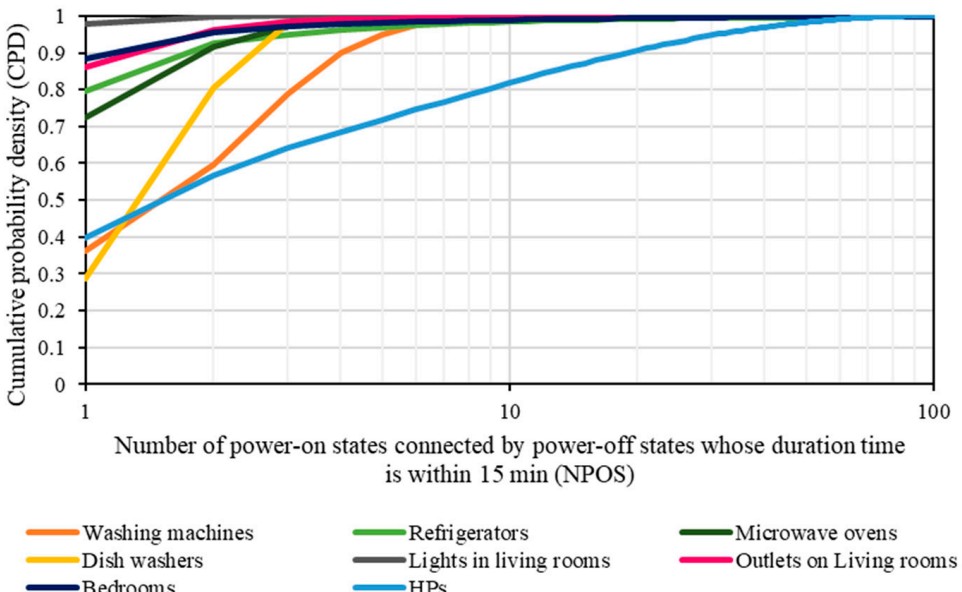

**Figure 7.** Cumulative probability density (CPD) of number of power-on states connected by power-off states, with time duration of ΔOFF within 15 min.

## 4. Proposed Method to Detect Turn-On/Off State of HPs

### 4.1. Outline of the Proposed Method

Based on the discussion in the previous section, we proposed a rule-based method for estimating the turn-on/off state of HPs from the monitored household total electricity consumption. This method involved two steps.

- Pre-process: Based on the pre-acquired total electricity demand data of a dwelling in the middle season, the time-series characteristics of the power consumption patterns of sequential operation appliances other than HPs were extracted. In addition, the power consumption baseline for the dwellings was determined. (Section 4.2).
- Detection: The electricity consumption corresponding to the sequential operation appliances, except for HPs, was subtracted from the measured total household electricity consumption (Section 4.3.1). Subsequently, the turn-on/off states of the HPs were estimated based on the duration of high-power consumption (Section 4.3.2) and time-series characteristics (Section 4.3.3).

For pre-processing, using the electricity data of the target dwelling would be ideal; however, if this is unavailable, data from dwellings with similar conditions can be used. This method can be adapted to dwellings with multiple HPs in which at least one HP is operating. Therefore, this method does not estimate the number of HPs during the operation.

### 4.2. Pre-Processing Based on Electricity Consumption Data in Middle Season

A flowchart of the pre-processing is shown in Figure 8. First, the average daily pattern of household electricity demand was calculated for the target dwelling based on the pre-acquired demand in the middle seasons with no HP use, which can be considered as the baseline demand. Subsequently, the household total power consumption patterns were characterised by a set of ($\Delta ON_i$, $\Delta t_{on,i}$, $\Delta OFF_i$) associated with each sequential operation appliance. In addition, the set of ($\Delta ON_i$, $\Delta t_{on,i}$, $\Delta OFF_i$) was examined to check whether the following three conditions were satisfied for all combinations of i and j.

$$(1-b) \cdot \Delta ON_j \ \leq \ \Delta ON_i \ \leq \ (1+b) \cdot \Delta ON_j \ \ \forall i,j \in \ I_{mid} \tag{8}$$

$$\Delta t_{on,j} - c \ \leq \ \Delta t_{on,i} \leq \ \Delta t_{on,j} + c \qquad \forall i,j \in \ I_{mid} \tag{9}$$

$$(1-b) \cdot \Delta OFF_j \ \leq \ \Delta OFF_i \ \leq \ (1+b) \cdot \Delta OFF_j \qquad \forall i,j \in \ I_{mid} \tag{10}$$

where i and j are indices of time step (i $\neq$ j), and $I_{mid}$ indicates the set of time steps for the electricity data in the middle season. The parameter b was tentatively set to 0.05 in this study considering the measurement accuracy of the Osaka dataset ($\pm$5%), and the parameter c was set to 1 min. This is because the dataset used in this study was measured in cumulative electricity consumption (Wh) per minute, and $\Delta t$ may deviate by $\pm$1 min from the actual start time of operation of appliances.

When these three conditions were satisfied, i and j were judged to have almost the same time-series characteristics and thus were likely to characterise the behaviours of the same sequential operation appliance. In such a case, ($\Delta ON_i$, $\Delta t_{on,i}$, $\Delta OFF_i$) was renamed ($\Delta \check{ON}_k$, $\Delta \check{t}_{on,k}$, $\Delta \check{OFF}_k$) and added to the 'power-on state pattern table', which stored the variables related to the signals of sequential operation appliances.

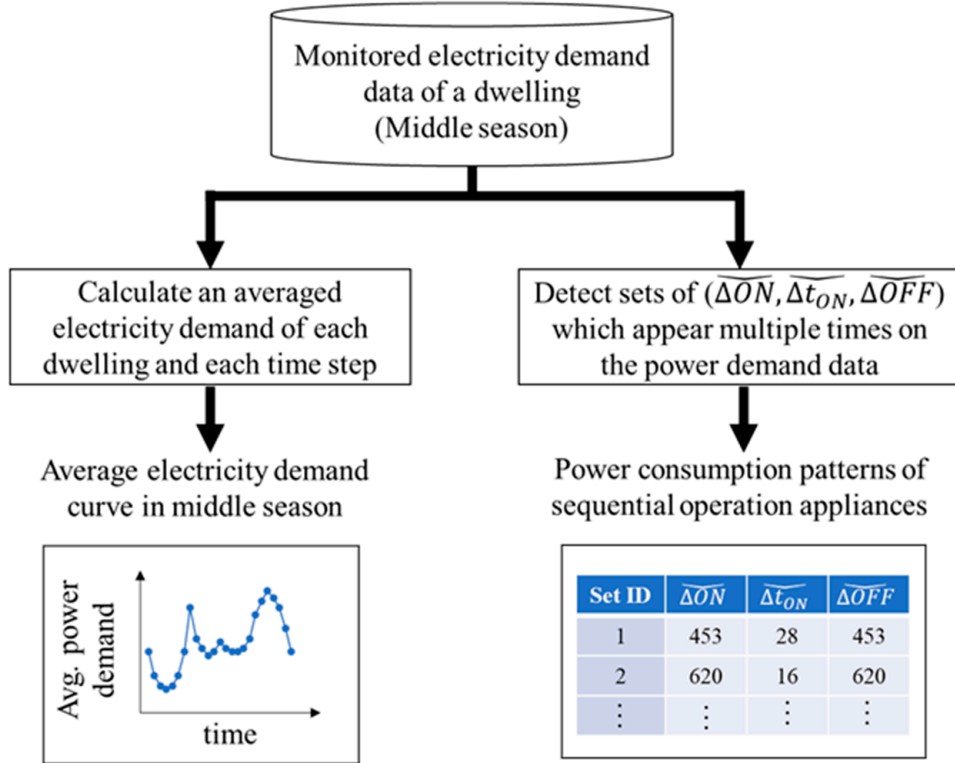

**Figure 8.** Flowchart of pre-processing.

*4.3. Detection of Turn-On/Off State of HPs*

4.3.1. Subtraction of Power Consumption of Sequential Operation Appliances except for HPs

The estimation process is shown in Figure 9. First, the time-series characteristics of ($\Delta ON_i$, $\Delta t_{on,i}$, $\Delta OFF_i$) were computed from the monitored household electricity P(t), and we checked whether the following three conditions were satisfied by referring to the power-on state pattern table created in the pre-process.

$$(1-b) \cdot \Delta \check{ON}_k \leq \Delta ON_i \leq (1+b) \cdot \Delta \check{ON}_k \qquad \forall i \in I_{tar}, k \in K \qquad (11)$$

$$\Delta \check{t}_{on,k} - c \leq \Delta t_{on,i} \leq \Delta \check{t}_{on,k} + c \qquad \forall i \in I_{tar}, k \in K \qquad (12)$$

$$(1-b) \cdot \Delta \check{OFF}_k \leq \Delta OFF_i \leq (1+b) \cdot \Delta \check{OFF}_k \qquad \forall i \in I_{tar}, k \in K \qquad (13)$$

where $I_{tar}$ indicates the set of time steps for the target demand data for estimating the turn-on/off state of the HPs, k indicates the identification number of variables stored in a power-on state pattern table, and K is the set of k. When all conditions (11)–(13) were satisfied, the total power consumption P(t) was subtracted from $\Delta ON_i$ for a period from t to $+\Delta t_{on,i}$ based on the assumption that a sequential operation appliance operates within $\Delta t_{on,i}$ with an electricity consumption of $\Delta ON_i$.

$$P(t) = P(t) - \Delta ON \qquad (14)$$

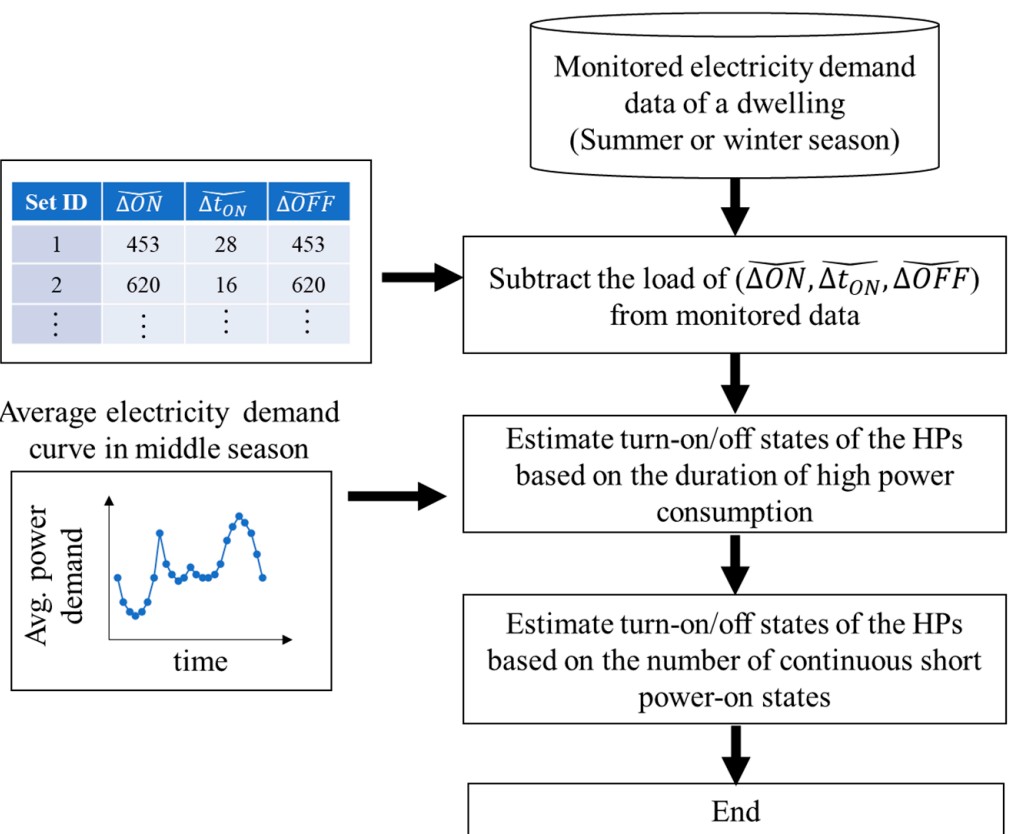

**Figure 9.** Overall flow of the estimation process.

$P(t)$, hereafter, represents the electricity consumption after removing the power consumption of the sequential operation appliances.

### 4.3.2. Detection Based on the Duration of High-Power Consumption

Processed electricity $P(t)$ was first assessed in terms of the duration of high-power consumption using Equation (15), as shown in Figure 10.

$$P(t) > P_t^{base} + \gamma \tag{15}$$

where $P_t^{base}$ refers to the baseline of the total electricity consumption at time step t calculated at the pre-processing. The time increment for $P_t^{base}$ does not have to be 1 min but can be larger, that is 1 h. $\gamma$ is a parameter with unit W. The value of $\gamma$ is assumed to be larger than the power consumption of low-power devices (in our dataset, lights in living rooms, refrigerators, and washing machines) and smaller than the power consumption of HPs. In this study, $\gamma$ was set to 150 W because $\Delta$ON of HPs was largely over 150 W, as shown in Figure 3.

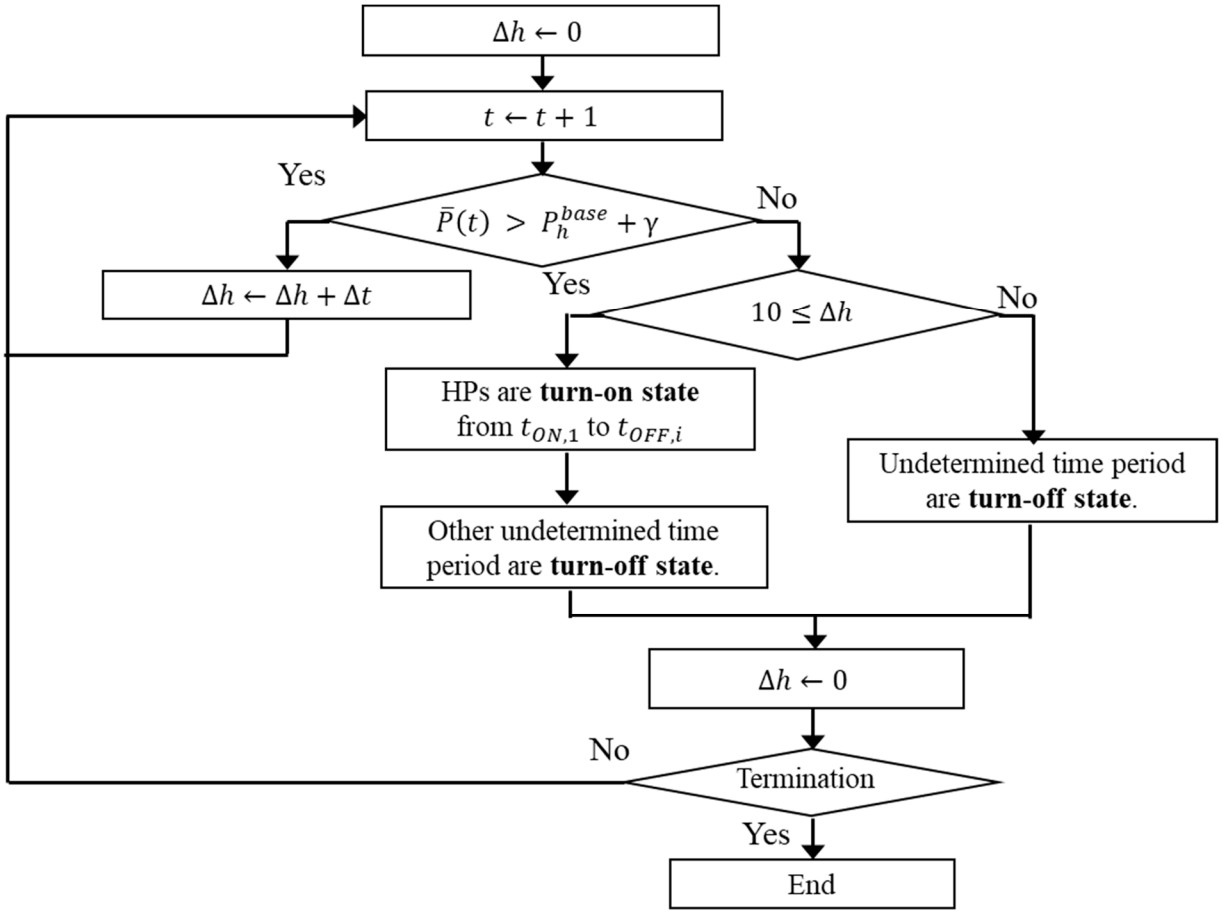

**Figure 10.** Estimation flow based on high-power duration.

When $P(t)$ satisfies Equation (15), $\Delta h$ is replaced by $\Delta h + \Delta t$ ($\Delta t$ is the time step interval that is 1 min in our dataset); otherwise, $\Delta h$ is replaced by zero. We assumed that an HP is in a turn-on state when $\Delta h$ is larger than a threshold. This threshold is set to 10 min in our study to avoid the misclassification of turn-on events of other appliances requiring high-power consumption in a very short period, such as microwave ovens.

### 4.3.3. Detection Based on Time-Series Characteristics

The processed electricity $P(t)$ was evaluated in terms of time-series characteristics, that is, the number of power-on states connected by short power-off states, for identifying HP operations with low and intermittent power consumption. The transition from the power-on state to the power-off state of an appliance was identified when the difference in the processed electricity $P(t)$ between the two time steps, namely $\Delta P(t)$, satisfied Equation (17). The transition from the power-off state to the power-on state was also identified when $\Delta P(t)$ satisfied Equation (18).

$$\Delta P(t) = P(t+1) - P(t-1) \tag{16}$$

$$150 \leq \Delta P(t) \leq 400 \tag{17}$$

$$-400 \leq \Delta P(t) \leq -150 \tag{18}$$

Using $\Delta P(t)$, $\Delta t_{ON,i}$ and $\Delta t_{OFF,i}$ are computed based on Equations (2)–(5). If more than three consecutive on/off events satisfied Equations (19) and (20), the sequence of these events was assumed to be HP-related.

$$\Delta t_{ON,i} \leq 15 \tag{19}$$

$$\Delta t_{OFF,i} \leq 15 \qquad (20)$$

Specifically, an HP is considered to be in the turn-on state from the time when the first power-on state starts ($t_{ON,i-A}$) to the time when the last power-on state is finished ($t_{OFF,i}$). However, when the value of A is less than 2, the HP is considered to be in the turn-off state, as other appliances may be operated simultaneously. This allows an accurate turn-on/off-state estimation for HPs, including inverter HPs (Figure 11).

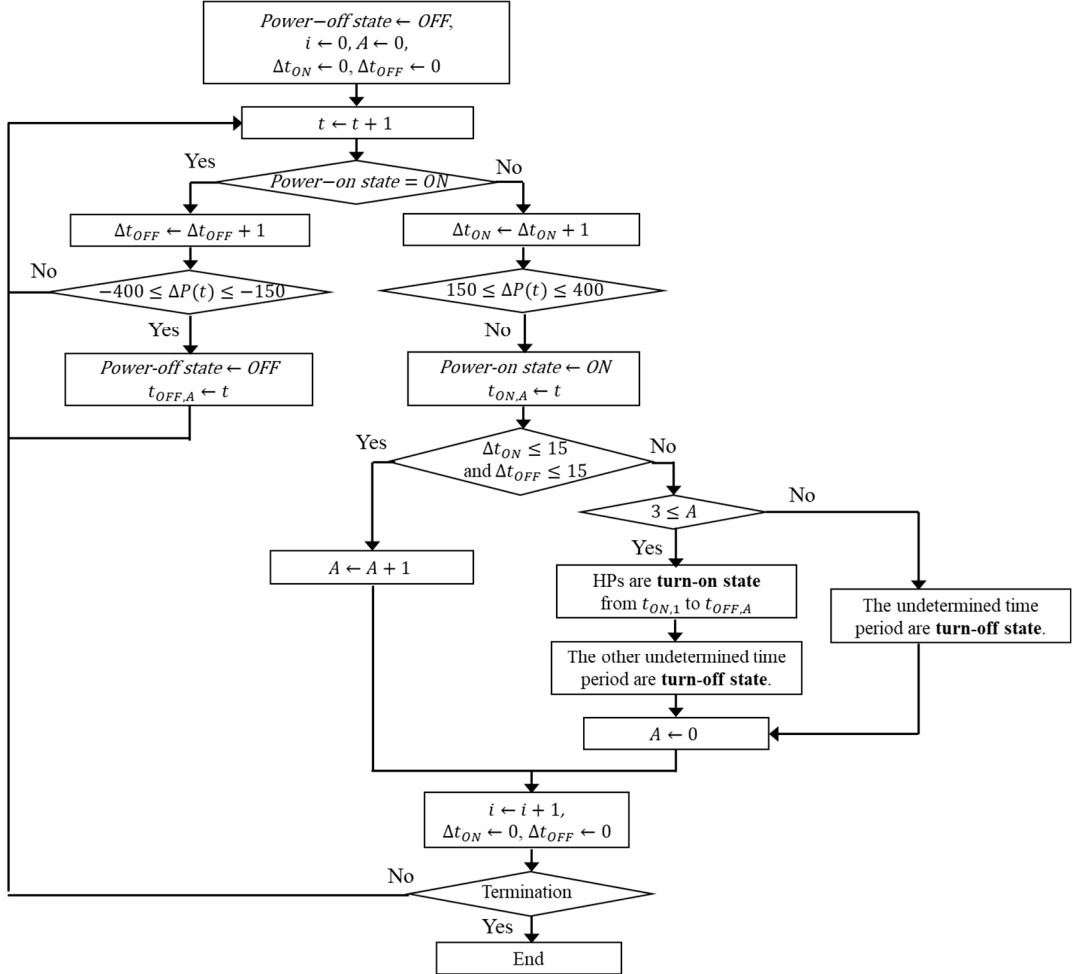

**Figure 11.** Estimation flow based on the time-series characteristics.

## 5. Validation of the Proposed Model

### 5.1. Data Used for Validation

The accuracy of the proposed method was assessed using measured data from a housing complex in Osaka, Japan (Section 2.1). From the dataset of the 586 dwellings, 423 dwellings were used that contained data with negligible error. The remaining 163 dwellings were excluded because of a significant error in their data. Additionally, the total household demand measured from 8 May to 24 May 2013 was used for pre-processing, and the household demand data measured from 1 June to 30 September 2013 were used to estimate the turn-on/off state of HPs. In addition, the occupants' intended turn-on/off states of the HPs were identified using the electricity consumption data of HPs by adopting the method proposed by Ono et al. [36]. The time increment for the baseline load $P_t^{base}$ was set as 1 h.

For comparison, we also applied the method proposed by Inoue et al. [25], which used the AODE. This model assumes that when high-power consumption continues for a certain duration, the HPs are likely to be in the turn-on state.

### 5.2. Time Sequence of Estimated On/Off State

Figure 12 shows two contrasting time sequences of the estimated turn-on/off states of the HPs based on the proposed model. Figure 12a shows the results of the duration with high HP power consumption, and Figure 12b shows the intermittent power-on/off states of the HPs. The black line represents $\Delta P(t)$ expressed by Equation (16), that is, the baseline plus 150 W.

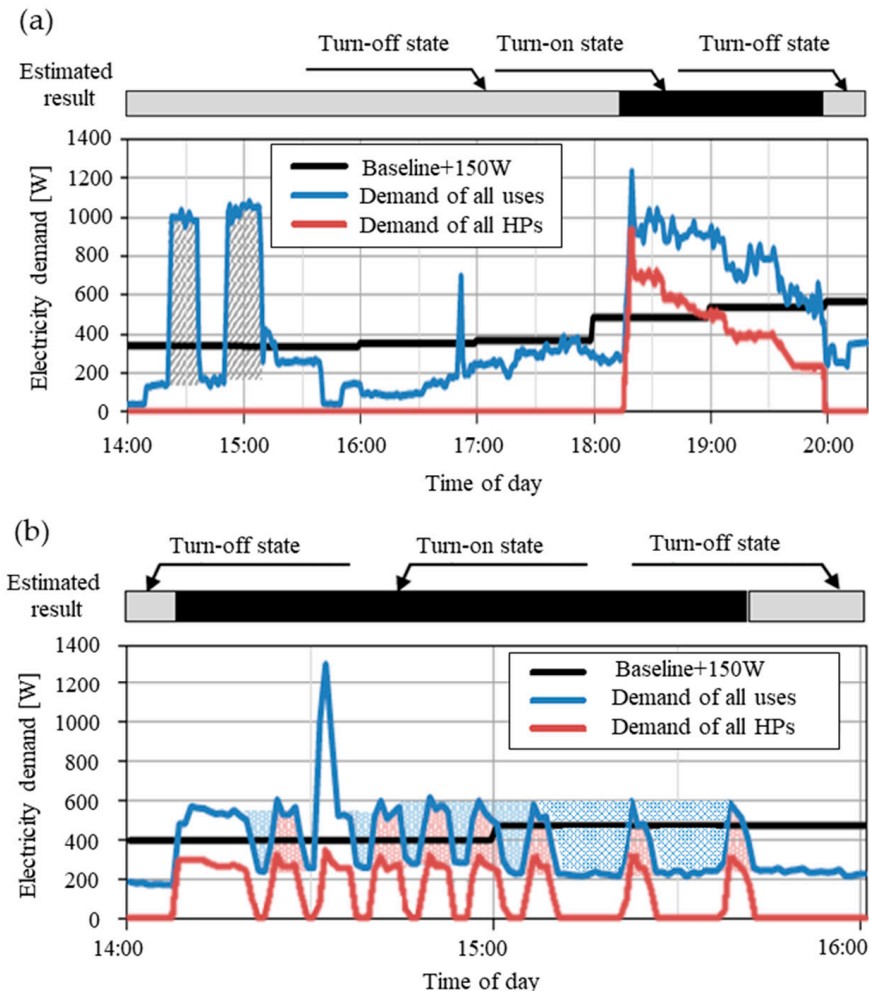

**Figure 12.** Time sequence of monitored electricity of household total and HP total and estimated turn-on/off state of HPs for duration with (**a**) a single turn-on state and (**b**) an intermittent HP operation.

In Figure 12a, the total household demand of all uses recorded approximately 1000 W twice between 14:20 h and 15:20 h for 15 to 20 min. The proposed model accurately identified these events as sequential operation appliances other than HPs; thus, the estimated HP state was turned off. Conversely, the time when the total household demand exceeded the baseline load between 18:10 h and 20:00 h was identified as the HP turn-on state.

In Figure 12b, the intermittent power-on/off states are marked by the red and blue hatched areas, respectively. When the time durations of the power-on/off states $\Delta t_{on}/\Delta t_{off}$ were less than 15 min and the differences in household total electricity demand $\Delta ON/\Delta OFF$ were between 150 W and 400 W, the proposed model judged the entire period of this intermittent power fluctuation as a continuous turn-on state operated by the occupants. The sharp increase in household demand of over 1000 W at approximately 14:32 h was due to the operation of a microwave oven. Overall, the comparison between the estimated HP states and the monitored data of these figures showed good correspondence.

### 5.3. Performance Metrics of the Proposed Model

The accuracy of the proposed method and AODE method [25] was evaluated using the precision, recall, and F-score determined by Equations (21)–(23).

$$\text{Precision} = \frac{\text{TP}}{\text{TP} + \text{FP}} \tag{21}$$

$$\text{Recall} = \frac{\text{TP}}{\text{TP} + \text{FN}} \tag{22}$$

$$\text{F} - \text{score} = \frac{2 \times \text{Precision} \times \text{Recall}}{\text{Precision} + \text{Recall}} \tag{23}$$

where TP is true positive, in which at least one HP is in the turn-on state, and the estimation is also in the turn-on state. FP is false positive, in which all HPs are in the turn-off state; however, the model judged that at least one HP was in the turn-on state. FN is a false negative in which at least one HP was in the turn-on state; however, the model judged that no HP was in the turn-on state.

The performance of the AODE method for the current energy data indicated an F-score of 0.667 and precision and recall of approximately 60–70%. Notably, Inoue et al. [25] reported an F-score of approximately 0.9 for dwellings with non-IHPs. The discrepancy in the F-scores between Inoue et al. [25] and Table 3 suggests the limitation of their model for detecting OBs using IHPs. In contrast, the proposed model outperformed the AODE method, with higher values for all the metrics.

**Table 3.** Estimation accuracy of the proposed model and AODE method.

|  | F-Score | Precision | Recall |
|---|---|---|---|
| Inoue et al. [25] | 0.667 | 0.708 | 0.708 |
| Proposed method | 0.834 | 0.820 | 0.847 |

Figure 13 depicts the relationships between the time ratio of the HP operation and accuracy indicators, namely precision, recall, and F-score. The time ratio of the HP operation indicates the total number of hours in which one or more HPs were switched on during the period divided by the targeted period. Each plot represents the data for each dwelling.

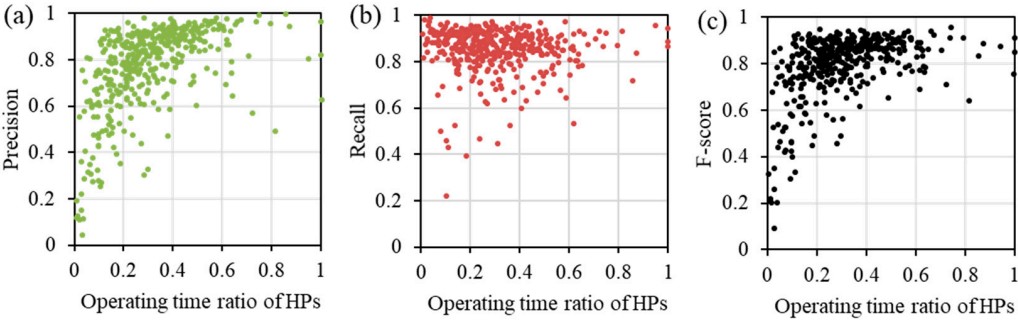

**Figure 13.** Relationship between operation time ratio of HPs and accuracy indicators of (**a**) precision, (**b**) recall and (**c**) F-score.

As shown in Figure 13a, the scattered plot indicates a weak positive relation, and most dwellings with a time ratio of over 0.4 show a precision of over 0.6. In contrast, dwellings with a time ratio below 0.2 tend to show low precisions of approximately 0.05. In these dwellings, occupants rarely used HPs; thus, the frequency of true positives (i.e., the model correctly judged as the turn-on state) is relatively small. Consequently, even a small number of false positives (i.e., the model incorrectly judged as the turn-on state when no HPs were running) resulted in poor ratings. In Figure 13b, most of the dwellings show a recall of over

0.8, regardless of the time ratio, and the reduction in the performance metrics for dwellings with a low time ratio is less evident compared to the precision results. In dwellings where HPs were rarely used, the frequency of true positives–the model correctly determining that HPs were in use–is lower, as mentioned above. However, the frequency of false positives–incorrectly assuming that HPs were in use–is not remarkably high. The trend of the scatter plot for the F-score (Figure 13c) is generally similar to that for the precision.

Figure 14 shows the histogram and CPD of the F-scores calculated for each dwelling. The F-scores for most dwellings ranged from 0.85 to 0.9, and 86% of the dwellings had F-scores above 0.7. Contrastingly, F-scores of remaining dwellings (14%) showed a long tail distribution with relatively low performance.

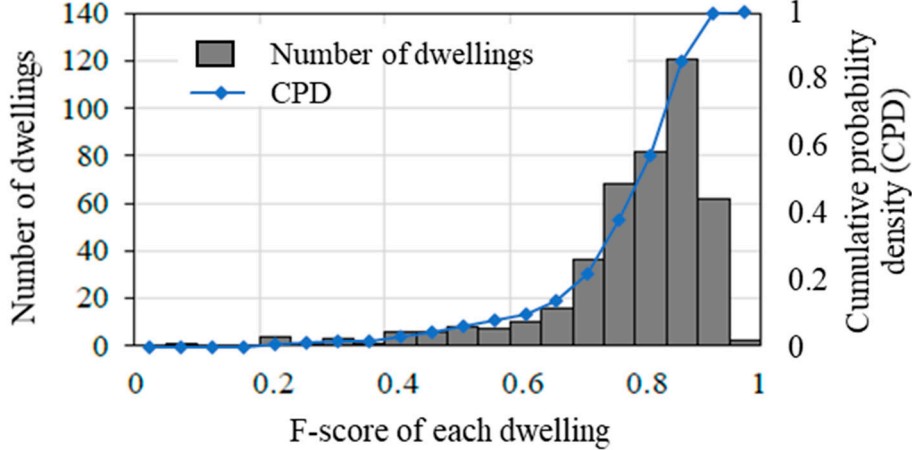

**Figure 14.** Distribution of F-scores computed for 423 dwellings.

Figure 15 shows the variations in estimation accuracy with month and time of the day. August and July 2013 were the hottest months, and June and September were relatively cooler at the site where the load data were collected. The F-score for July and August was approximately 0.9, whereas that for September and June was less than 0.75. Precision and recall also exhibited a similar tendency; however, it was more evident for recall. This indicates that during the few months when HPs were not frequently used, the number of events in which they were mistakenly judged to be in the turn-on state increased.

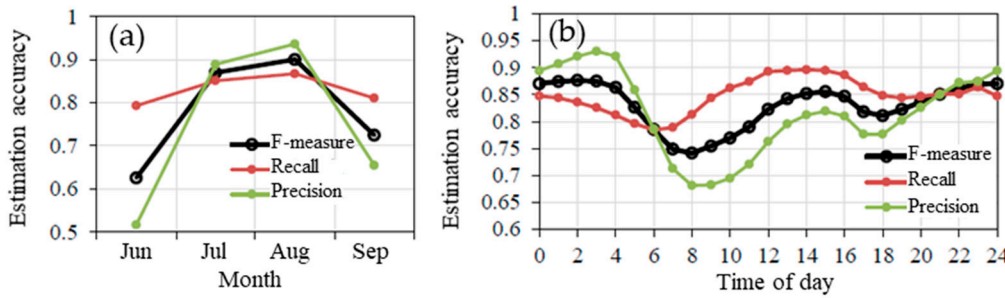

**Figure 15.** Influence of (**a**) season and (**b**) time of day on estimation accuracy.

With regard to the influence of the time of day on the accuracy, the F-score and precision exhibited similar trends of highest accuracy at around 3:00 h and lower accuracy at around 8:00 h. The recall was the maximum from 12:00 to 16:00 h and the minimum at 6:00 h in the morning, which differed from the other two indicators. This is probably due to the high outdoor temperature from noon to evening, resulting in a large cooling load and high-power consumption of HPs. On the contrary, the temperature around 6:00 h was the lowest throughout the day, and the air-conditioning load was small, making it difficult to detect signals related to HP electricity. In addition, the daily variations in precision were larger than those of the other two indicators. This is because precision is more sensitive to

how long the HPs are used compared to recall, as shown in Figure 15. The precision with high values from 0:00 to 4:00 h and from 14:00 to 16:00 h and low values at approximately 8:00 and 18:00 h. This is consistent with the high usage frequency of home appliances, except for HPs at night and in the morning, which tends to increase the false detection of the HP operation state.

## 6. Conclusions

To effectively promote behavioural changes for energy saving and DR, understanding the characteristics of OBs related to HPs in the residential sector based on data is crucial. However, in the large-scale worldwide deployment of smart metres in residences, only the total household electricity is measured, and data on the power consumption of individual home appliances are rarely obtained. This study proposed a rule-based method to identify the OBs related to air conditioning in dwellings with IHPs from the time-series total household electricity data. The main findings are summarised as follows:

- The power consumption patterns of HPs currently used in many countries are characterised by intermittent power-on/off operation, depending on the intensity of the room heating and cooling load, owing to the inverter control. To quantify such features of IHPs different from other home appliances, three indicators ($\Delta ON_i$, $\Delta t_{on,i}$, $\Delta OFF_i$) are defined.
- The proposed method entails two steps: (1) pre-processing to determine the baseline demand for the mid-season and extract time-series characteristics of the power consumption patterns of sequential operation appliances, and (2) a detection process using the baseline demand and the abovementioned three indicators.
- The performance of the proposed model was validated using data measured from 423 dwellings. The F-scores, precision, and recall showed performance better than those in the previous study.

**Author Contributions:** Conceptualisation, T.O; methodology, T.O.; data analysis, T.O.; investigation, T.O.; writing—original draft preparation, T.O.; writing—review and editing, A.H.; visualisation, T.O.; supervision, J.T.; project administration, A.H.; funding acquisition, A.H. All authors have read and agreed to the published version of the manuscript.

**Funding:** This research was funded by JSPS KAKENHI (Grant Number 15K06324).

**Institutional Review Board Statement:** Not applicable.

**Informed Consent Statement:** Not applicable.

**Data Availability Statement:** Not applicable.

**Acknowledgments:** The authors wish to thank general incorporated foundation low-carbon city development corporation.

**Conflicts of Interest:** The authors declare no conflict of interest.

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
