# Peer review of "Non-Intrusive Detection of Occupants’ On/Off Behaviours of Residential Air Conditioning"

_sustainability, doi:10.3390/su142214863_

Round 1

Reviewer 1 Report

This article proposed a rule-based method for identifying the occupants’ intended operation states of IHPs based on the statistical analysis of load data monitored at 423 dwellings. It is believed this method can provide insight understanding of individual air conditioning equipment’s status thus help to optimize the energy consumption. The finding is well presented and analyzed, the following comments are to help to improve the quality of this paper:

1.       Although the real consumption data was used, there is still lack of labeled data for this training-based method. It is very essential to at least have a certain amount of labeled data in order to make a solid conclusion about the improved estimation accuracy.

2.       The purpose or future application of this method is not very clearly described, it is suggested to add some contents to explain how the outcomes will be utilized by utility companies etc.

Author Response

We appreciate to the reviewer for your kind review of our submitted paper despite your busy schedules. We have summarized our responses to the referee review in the attached pdf file.

Reviewer 2 Report

Comments for Sustainability-1927956

The study proposed a rule-based method to identify the OBs related to air conditioning at dwellings with IHPs from the time-series total household electricity data. Here are some suggestions.

1.     The full name should be indicated before abbreviation is used, e.g. in the abstract only the full name of HPs is stated but the full name of IHPs is not mentioned.

2.     Language needs to be polished by professional experts who are experienced with technical English writing.

3.     Please consider a diagram to more clearly represent the research process and overall ideas.

4.     The discussion of results is weak and it is recommended to include further explanation of the limitations of this work as well as future work.

5.     The paper mentions: However, few studies have considered such characteristics for detecting IHP operations. Please point out the differences between this work and “few studies” that considers the characteristics for detecting IHP operations in order to highlight the innovation of this paper.

Author Response

(The authors gave the same response as above.)

Round 2

Reviewer 2 Report

The revision respond all the comments. 

Here are some additional comments. 

1)Total 586 dwellings in table 1. But abstract presents 423 dwellings. Check other figures to enure the consistance. 

2)Please check the language of the paper. Better to have native speaker to proofread.  

Author Response

We appreciate to the reviewer for your careful review of our submitted paper despite your busy schedules. We have summarized our responses to the referee review as attached word file.
